

# Clinical considerations on antimicrobial resistance potential of complex microbiological samples

Norbert Solymosi[1,2], Adrienn Gréta Tóth[1,2], Sára Ágnes Nagy[2],
István Csabai[2], Csongor Feczkó[1], Tamás Reibling[1] and Tibor Németh[3]

[1] Centre for Bioinformatics, University of Veterinary Medicine, Budapest, Hungary
[2] Department of Physics of Complex Systems, Eötvös Loránd University, Budapest, Hungary
[3] Department and Clinic of Surgery and Ophthalmology, University of Veterinary Medicine, Budapest, Hungary

## ABSTRACT

Antimicrobial resistance (AMR) is one of our greatest public health challenges. Targeted use of antibiotics (ABs) can reduce the occurrence and spread of AMR and boost the effectiveness of treatment. This requires knowledge of the AB susceptibility of the pathogens involved in the disease. Therapeutic recommendations based on classical AB susceptibility testing (AST) are based on the analysis of only a fraction of the bacteria present in the disease process. Next and third generation sequencing technologies allow the identification of antimicrobial resistance genes (ARGs) present in a bacterial community. Using this metagenomic approach, we can map the antimicrobial resistance potential (AMRP) of a complex, multi-bacterial microbial sample. To understand the interpretiveness of AMRP, the concordance between phenotypic AMR properties and ARGs was investigated by analyzing data from 574 *Escherichia coli* strains of five different studies. The overall results show that for 44% of the studied ABs, phenotypically resistant strains are genotypically associated with a 90% probability of resistance, while for 92% of the ABs, the phenotypically susceptible strains are genotypically susceptible with a 90% probability. ARG detection showed a phenotypic prediction with at least 90% confidence in 67% of ABs. The probability of detecting a phenotypically susceptible strain as resistant based on genotype is below 5% for 92% of ABs. While the probability of detecting a phenotypically resistant strain as susceptible based on genotype is below 5% for 44% of ABs. We can assume that these strain-by-strain concordance results are also true for bacteria in complex microbial samples, and conclude that AMRP obtained from metagenomic ARG analysis can help choose efficient ABs. This is illustrated using AMRP by a canine external otitis sample.

Corresponding author
Norbert Solymosi,
solymosi.norbert@gmail.com

# GRAPHICAL ABSTRACT

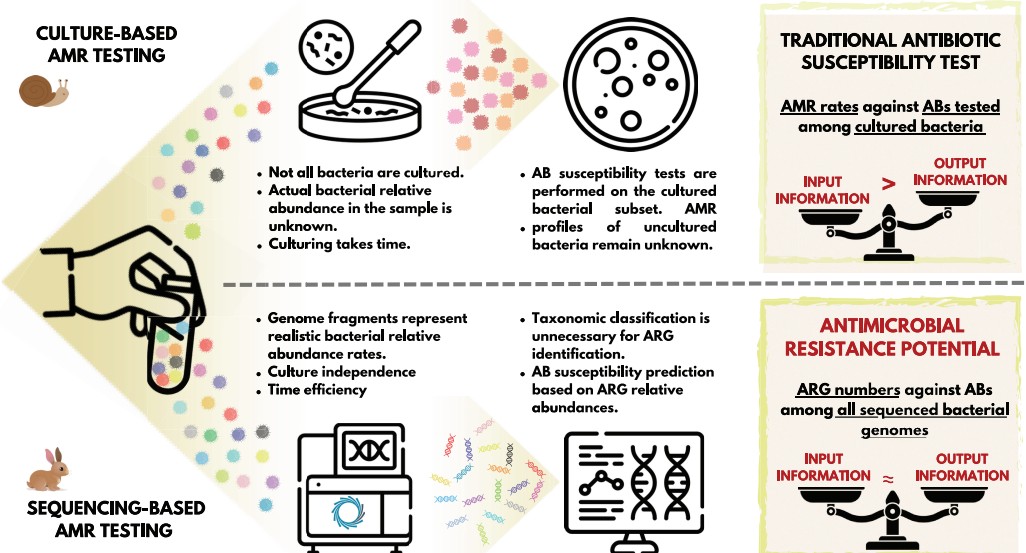

# INTRODUCTION

The appearance and use of antibiotics (ABs) has brought an unprecedented advance in the treatment of bacterial infections (*Muteeb et al., 2023*; *Solymosi et al., 2024*). The general consensus is that the case-specific, targeted use of ABs is favorable in clinical settings. This means using AB compounds that are supposed to be effective against the bacteria involved in the disease process to be treated (*Magnusson, 2020*). The classic standard is to take a sample from the patient for microbiological testing, and perform the bacterial culturing and the AB susceptibility testing (AST) of it (*Motro & Moran-Gilad, 2017*). Based on these tests, we obtain information regarding the set of ABs that the bacteria in the culture showed susceptibility to, and the set of compounds to which they are moderately or not at all susceptible (*Khan, Siddiqui & Park, 2019*). The general procedure for this widely-used routine bacteriology technique is to first obtain a mixed culture from the sample, then select certain colonies for pure culture (*Boolchandani, D'Souza & Dantas, 2019*), and perform disk diffusion or microbroth dilution AST (*Boolchandani, D'Souza & Dantas, 2019*). Importantly, even in a mixed bacterial culture, only a subset of the microorganisms and bacteria involved in the pathological process are represented. One partial explanation for this is that a significant proportion of the bacteria that have been identified to date are challenging to culture (*Steen et al., 2019*). Furthermore, even in case of cultivable species, providing the same culture conditions (*e.g.*, type of medium) favors the growth of some taxa over others (*Stewart, 2012*). The latter issue can be avoided by creating mixed cultures in parallel, providing different culture environments (*Van Belkum et al., 2013*). Nevertheless, AB susceptibility testing is performed on an unpredictably small subset of bacterial strains. Additionally, several phases of traditional microbiological testing are time-consuming processes that take at least a few days to complete. Furthermore, the entire conventional diagnostic workflow is rather labour-intensive (*Motro & Moran-Gilad, 2017*).

At the end of the diagnostic process, the phenotypic AMR profile delivered to clinicians refers to a fraction of the bacteria present in the disease process.

At the same time, next-generation sequencing facilitates the identification of a wide variety of genes, including antimicrobial resistance genes (ARGs). Moreover, the identification of the ARGs in microbiological samples can be performed in a considerably shorter time, regardless of whether the bacteria carrying them have been cultured or not yet, or whether they are considered pathogenic or not (*Motro & Moran-Gilad, 2017*). While the classical approach outlined above yields a phenotypic AB susceptibility profile, ARG detection yields a genotypic AMR profile (*Boolchandani, D'Souza & Dantas, 2019*). The AB resistance detected in phenotypic settings reflect the activity and functionality of the genetic background (*Chibucos et al., 2014*), whether or not the genes or polymorphisms that contribute to the generation of the expressed effect, AMR, are known. In the case of a genotypic resistance profile, we do not know whether the gene is expressed or not. However, we can assume that there is a possibility for the expression of the ARGs present that can be interpreted as antimicrobial resistance potential (AMRP).

For the clinical evaluation of AMRP, it is essential to know the extent to which phenotypic AMR equals with the presence or absence of ARGs. As outlined above, the assessment of this from complex microbiological samples is significantly biased. However, by examining pure bacterial cultures, we can get an idea of the correspondence between AMRP and phenotypic AMR. If AMRP and phenotypic AMR exhibit a high degree of overlap for a given set of strains, it can be reasonably assumed that culturing all strains in a complex microbiological sample would yield comparable results. In such cases, the metagenomic ARG analysis of these complex samples may result in AMRP profiles that could contribute to the selection of ABs in the following way. Based on the AMRP, those ABs could be prioritised that are potentially impeded with the lowest probability by the ARG-harbouring bacteria involved in the pathogenesis. Following this approach, the concordance of the AMR phenotype with the presence of ARGs was initially analysed in 574 strains of *Escherichia coli* from five distinct BioProjects. Subsequently, a metagenomic ARG analysis was conducted on a complex microbiological sample from a case of chronic, canine otitis externa, with the objective of describing the AMRP of the complex microbiological sample.

## METHODS

Portions of this text were previously published as part of a preprint (https://www.biorxiv.org/content/10.1101/2024.05.14.594174v1). To investigate phenotypic and genotypic AMR, antibiograms and Illumina sequencing data of *E. coli* strains used by *Pataki et al. (2020)* were downloaded from the European Nucleotide Archive (ENA) at EMBL-EBI repository (https://www.ebi.ac.uk/ena/browser/home, accessed on 2/5/2024). Of these *E. coli* datasets, BioProjects that contained at least 10 samples and an antibiogram involving more than one AB compound were included in the analyses. In the antibiograms, the resistant (R), intermediate (I), susceptible (S) categories for each AB were defined by the authors publishing the datasets based on different AST standards and cut-off versions in each BioProject: PRJDB7087 (European Committee on Antimicrobial

Susceptibility Testing, EUCAST, 2018, number of records with R phenotype: 1154, I: 143, S: 1667), PRJEB14086 (EUCAST, 2013/652/EU, R: 231, I: 0, S: 467), PRJEB21880 EUCAST, 2015, R: 92, I: 0, S: 1294), PRJEB21997 (EUCAST, 2017, R: 806, I: 0, S: 748), PRJNA266657 (Clinical and Laboratory Standards Institute, CLSI, 2015, R: 245, I: 97, S: 570). Raw FASTQ files were obtained from each BioProject. Bioinformatic analyis of the FASTQ files began with quality control, followed by the trimming and filtering of the raw short reads usingTrimGalore (v.0.6.6, https://github.com/FelixKrueger/TrimGalore), setting a quality threshold of 20. More than 50 bp long reads were assembled to contigs by MEGAHIT (v1.2.9) (*Li et al., 2015*) using default settings.

The metagenomic sample was collected from the right external ear canal of a 4-year-old neutered male Belgian griffon before total ear canal ablation (TECA) or lateral bulla osteotomy (LBO) surgery. The dog was previously diagnosed with an aural mass submitted as an inflammatory polyp and end-stage external otitis. The detection of the symptoms took place around 6 weeks before the surgery. Multiple conservative treatment rounds with AB and anti-inflammatory agents had been performed previously with no success. No tumor tissues were identified *via* histopathology examination. The history of the dog did not include severe allergies. The skin of the ear canal was red, swollen, scaled, thickened, and covered with bloody, purulent ear discharge before the surgery. By the routine bacteriology testing *Pseudomonas aeruginosa* and, in small amounts, *Malassezia pachydermatis* were detected from the sample. Even though, the identified *Pseudomonas aeruginosa* strain appeared to be multi-resistant, ceftazidime, gentamicin, tobramycin, ciprofloxacin, marbofloxacin and polymyxin B were indicated to be potentially effective, if administered locally, in great concentrations. DNA extraction was performed with QIAamp PowerFecal Pro DNA Kit from Qiagen (Hilden, Germany) according to the manufacturer's instructions. The concentrations of the extracted DNA solutions were evaluated with an Invitrogen Qubit 4 Fluorometer using the Qubit dsDNA HS (High Sensitivity) Assay Kit. The metagenomic long-read library was prepared by the Rapid Barcoding Kit 24 V14 (SQK-RBK114.24) from Oxford Nanopore Technologies (ONT). The sequencing was implemented with a MinION Mk1C sequencer using an R10.4.1 flow cell from ONT. The basecalling was performed using dorado (https://github.com/nanoporetech/dorado, v0.4.3) with model `dna_r10.4.1_e8.2_400bps_fast@v4.2.0`, based on the POD5 files converted from the FAST5 files generated by the sequencer. The raw reads were adapter trimmed and quality-based filtered by Porechop (v0.2.4, https://github.com/rrwick/Porechop) and Nanofilt (v2.6.0) (*De Coster et al., 2018*), respectively.

For the Illumina short read samples, the ARG content of the sequence in the contigs and for the ONT metagenome sample in the long reads was identified using ResFinder (v4.5.0) (minimum coverage 60%, minimum identity 90%) (*Bortolaia et al., 2020*).

In the agreement analysis between phenotypic AMR and AMR prediction based on ARG detection, we only used AST results that did not use AB combinations, as the genetic background of resistance to single ABs is better understood than ARG determination of resistance to AB combinations. From AST results, only resistant and susceptible results were used. As only two of the five BioProjects in the study included I AST results, only S and R AST data were used to present the analyses in a consistent manner. However, the

BioProjects that also included I results were re-evaluated separately. For these two BioProjects, the non-susceptible (I + R) phenotype was used in addition to the susceptible (S) phenotype. ABs negatively affected by the detected ARGs were recorded and collected by each sample.

Based on the AST and ARG results, 2 × 2 cross-tabulations (*TP*: true positive, *TN*: true negative, *FP*: false positive, *FN*: false negative) were generated by each AB compound. These were used to calculate metrics to describe the predictive goodness of the genome-based prediction of the phenotype. Phenotypic AST results were the reference (ground truth). Accordingly, the *TP* group refers to strains with both phenotypic and genotypic resistance to a given AB compound, *TN* to strains with both phenotypic and genotypic sensitivity, *FP* to strains with phenotypic sensitivity and genotypic resistance, and *FN* to strains with phenotypic resistance without the detection of any ARGs to a given drug. In addition to sensitivity ($SE = TP/(TP + FN)$), specificity ($SP = TN/(TN + FP)$), negative predictive value ($NPV = TN/(TN + FN)$) and positive predictive value ($PPV = TP/(TP + FP)$), point and 95% CI estimates were also calculated for false resistance (major error rate, $ME = FP/(TN + FP)$) and false susceptibility (very major rate, $VME = FN/(TP + FN)$).

Internationally accepted guidelines and standards are also available to assess the accuracy of the different AST methods in relation to standard reference methods, *i.e.*, to determine *VME* and *ME* values. In the United States, the FDA (Food and Drug Administration) regulations for the marketing of different AB susceptibility testing devices are quite strict. Thus, the *ME* of the test device examined must not exceed 3% when testing the efficacy of different bacterial species-drug combinations. In case of *VME*, the proposed statistical criteria include an upper 95% confidence limit for the *VME* of 7.5% at most, and a lower 95% confidence limit for the *VME* of 1.5% at most (*FDA, US, 2009*). An international standard has also been established for the evaluation of AB susceptibility testing devices, which proposes similar but not identical criteria as acceptable limits of accuracy (*International Organization for Standardization, 2006*). The above guidelines and standards have been tailored to compare different phenotypic techniques rather than phenotypic and genotypic methods, which have the potential for greater variation. Thus, we have used the 5% margin of error used in statistics.

## RESULTS

In Figs. 1 and 2 we summarize the point and 95% CI estimates of the agreement, prediction goodness metrics for the phenotypic (ground truth) and genotype-predicted AMR, by ABs and BioProjects (*Solymosi et al., 2024*).

ABs for which the *NPV* was estimable and reached 90% in at least half of the BioProjects ($n = 21$): amikacin, azithromycin, ceftazidime, chloramphenicol, ciprofloxacin, colistin, ertapenem, fosfomycin, gentamicin, imipenem, kanamycin, meropenem, nalidixic acid, streptomycin, sulfamethoxazole, sulfisoxazole, temocillin, tetracycline, tigecycline, tobramycin, and trimethoprim. ABs for which the *PPV* was estimable and reached 90% in at least half of the BioProjects ($n = 15$): ampicillin, cefepime, cefotaxime, cefoxitin, chloramphenicol, ciprofloxacin, colistin, fosfomycin, gentamicin, kanamycin, piperacillin,

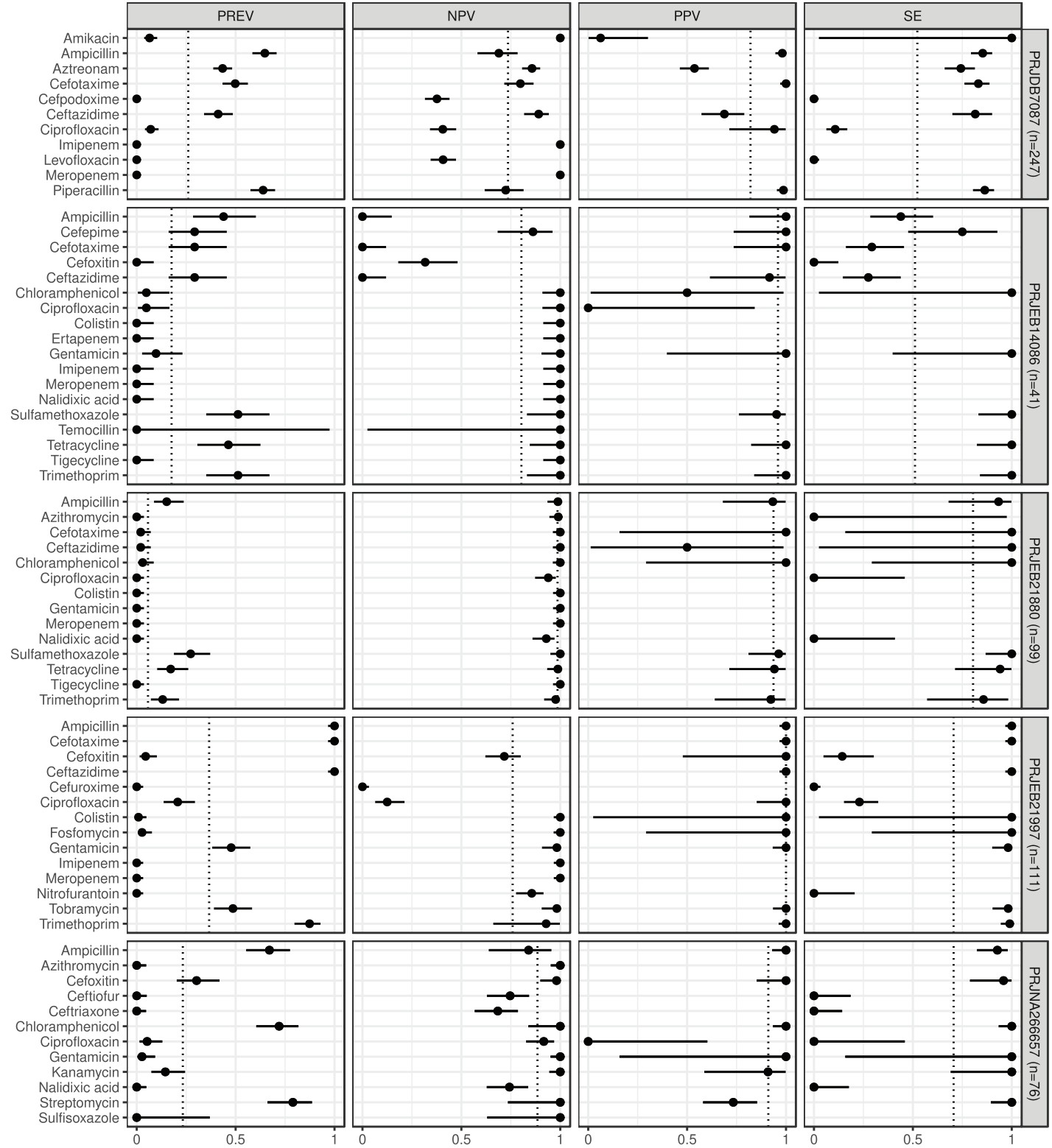

**Figure 1 Predictive performance by BioProjects.** Antimicrobial resistance gene-based prediction compared to the AB susceptibility test results using five BioProject datasets. Column PREV shows the prevalence (with 95% CI) of phenotypic antimicrobial resistance against certain ABs within the BioProjects. Point and 95% CI estimates for negative and positive predictive values, and sensitivity are presented in columns *NPV*, *PPV*, and *SE*. The dotted vertical lines represent the mean of the metrics within the given BioProject.

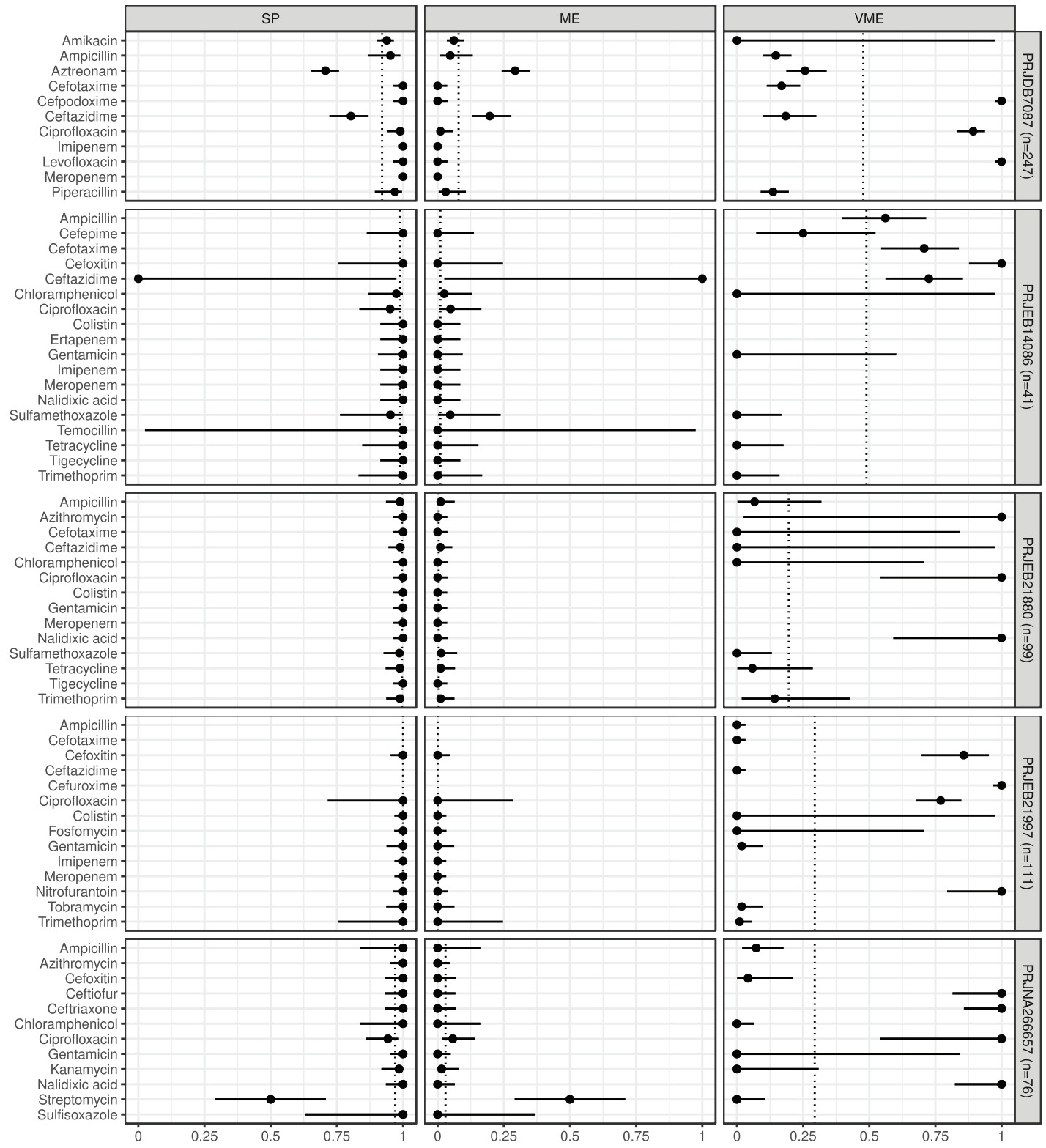

**Figure 2 Predictive performance by BioProjects.** Antimicrobial resistance gene-based prediction compared to the AB susceptibility test results using five BioProject datasets. Point and 95% CI estimates for specificity, major and very major error rates are presented in columns *SP*, *ME*, and *VME*. The dotted vertical lines represent the mean of the metrics within the given BioProject.

sulfamethoxazole, tetracycline, tobramycin, and trimethoprim. ABs for which the *SE* was estimable and reached 90% in at least half of the BioProjects ($n = 13$): amikacin, ampicillin, ceftazidime, chloramphenicol, colistin, fosfomycin, gentamicin, kanamycin, streptomycin, sulfamethoxazole, tetracycline, tobramycin, and trimethoprim. ABs for which the *SP* was estimable and reached 90% in at least half of the BioProjects ($n = 29$): amikacin, ampicillin, azithromycin, cefepime, cefotaxime, cefoxitin, cefpodoxime, ceftiofur, ceftriaxone, chloramphenicol, ciprofloxacin, colistin, ertapenem, fosfomycin, gentamicin, imipenem, kanamycin, levofloxacin, meropenem, nalidixic acid, nitrofurantoin, piperacillin, sulfamethoxazole, sulfisoxazole, temocillin, tetracycline, tigecycline, tobramycin, and trimethoprim. ABs for which *ME* was estimable and did not exceed 5% in at least half of the BioProjects ($n = 28$): ampicillin, azithromycin, cefepime, cefotaxime, cefoxitin, cefpodoxime, ceftiofur, ceftriaxone, chloramphenicol, ciprofloxacin, colistin, ertapenem, fosfomycin, gentamicin, imipenem, kanamycin, levofloxacin, meropenem, nalidixic acid, nitrofurantoin, piperacillin, sulfamethoxazole, sulfisoxazole, temocillin, tetracycline, tigecycline, tobramycin, and trimethoprim. ABs for which *VME* was estimable and did not exceed 5% in at least half of the BioProjects ($n = 15$): amikacin, cefotaxime, ceftazidime, chloramphenicol, colistin, fosfomycin, gentamicin, kanamycin, nalidixic acid, nitrofurantoin, streptomycin, sulfamethoxazole, tetracycline, tobramycin, and trimethoprim. Figure S1 shows the same metrics for the two BioProjects that also included the I AST phenotype.

The estimated metrics for ABs for which at least two BioProjects had identical cut-offs are shown in Fig. 3. According to these overall estimations the proportion of the ABs for which the *NPV* was estimable and reached 90% was 67% ($n = 8$): chloramphenicol, ciprofloxacin b, colistin, gentamicin, imipenem, meropenem, tigecycline, and trimethoprim. The proportion of the ABs for which the *PPV* was estimable and reached 90% was 67% ($n = 6$): ampicillin, cefotaxime, ciprofloxacin a, colistin, gentamicin, and trimethoprim. The proportion of the ABs for which the *SE* was estimable and reached 90% was 44% ($n = 4$): chloramphenicol, colistin, gentamicin, and trimethoprim. The proportion of the ABs for which the *SP* was estimable and reached 90% was 92% ($n = 11$): ampicillin, cefotaxime, chloramphenicol, ciprofloxacin a, ciprofloxacin b, colistin, gentamicin, imipenem, meropenem, tigecycline, and trimethoprim. The proportion of the ABs for which *ME* was estimable and did not exceed 5% was 92% ($n = 11$): ampicillin, cefotaxime, chloramphenicol, ciprofloxacin a, ciprofloxacin b, colistin, gentamicin, imipenem, meropenem, tigecycline, and trimethoprim. The proportion of the ABs for which *VME* was estimable and did not exceed 5% was 44% ($n = 4$): chloramphenicol, colistin, gentamicin, and trimethoprim.

The ARG frequencies associated with the AB identified in the metagenome analysis are summarised in Fig. 4, while Fig. 5 shows the ABs for which no ARG was detected in the sample (*Solymosi et al., 2024*).

## DISCUSSION

Based on the phenotypic and genotypic resistance concordance analyses in *E. coli* strains, the following conclusions can be drawn (*Solymosi et al., 2024*). For 39% of ABs, sensitivity

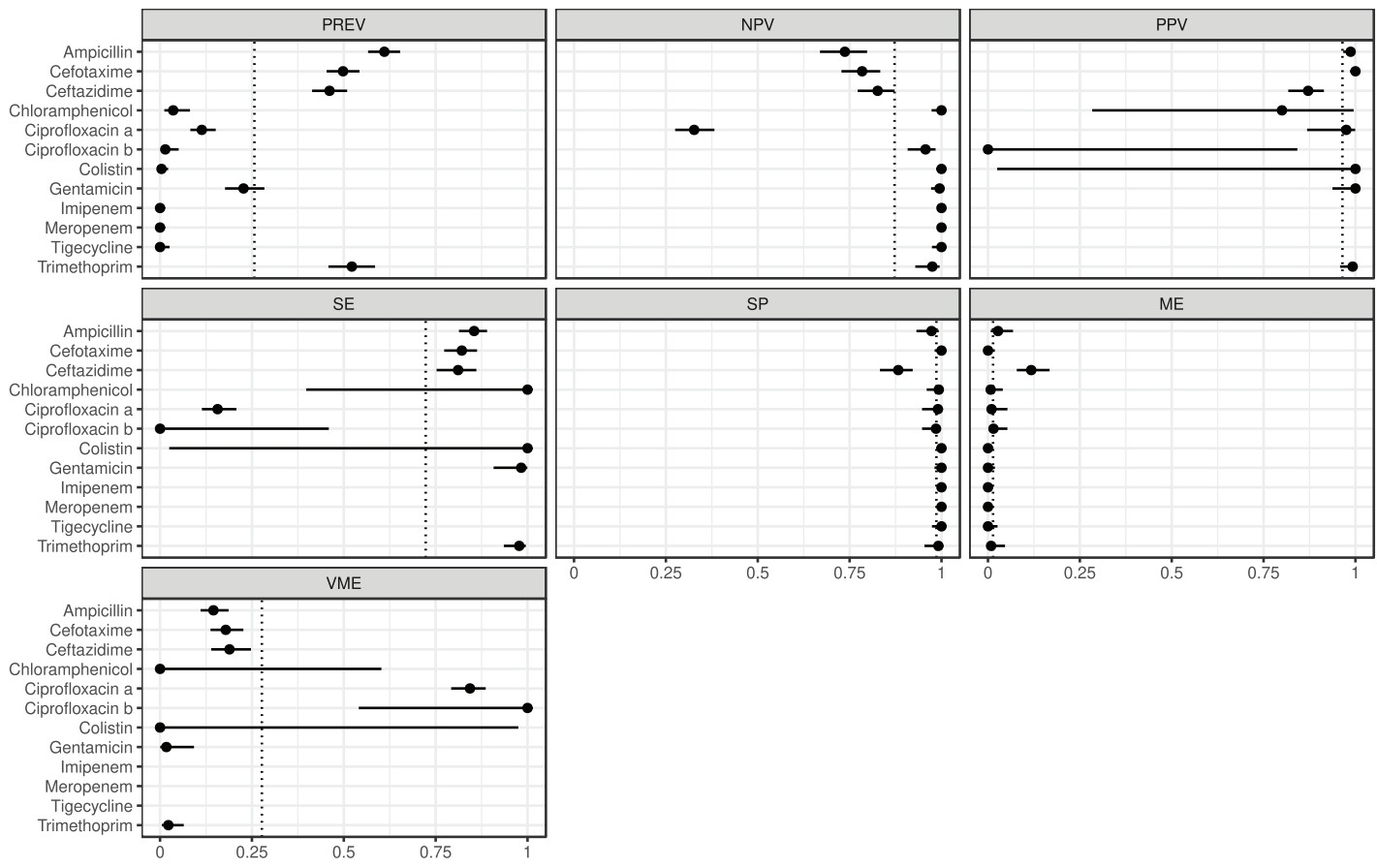

**Figure 3 Overall predictive performance for ABs having the same minimum inhibitory concentration (MIC) breakpoints in at least two BioProjects.** Column PREV shows the prevalence (with 95% CI) of phenotypic antimicrobial resistance against certain ABs within the BioProjects. Point and 95% CI estimates for negative and positive predictive values, sensitivity, specificity, major and very major error rates are presented in columns *NPV*, *PPV*, *SE*, *SP*, *ME*, and *VME*, respectively. In the case of ciprofloxacin there were two MIC breakpoint sets having data from two BioProjects. For ciprofloxacin a the MIC breakpoints were $S \leq 0.25$ and $R > 0.5$ mg/L, while for ciprofloxacin b $S \leq 0.5$ and $R > 1$ mg/L.

exceeded 90% in the evaluated BioProjects. In other words, for these ABs, phenotypically resistant strains are genotypically associated with a 90% probability of resistance. Based on the positive predictive values, a strain identified as genotypically resistant has at least a 90% probability of being phenotypically resistant for 45% of ABs. At 45% of ABs, there is less than a 5% probability that a phenotypically resistant strain will be identified as genotypically susceptible. By 88% of ABs, phenotypically susceptible strains are at least 90% likely to be susceptible based on genotype. Strains detected as genotypically sensitive are at least 90% likely to be phenotypically sensitive at 64% of the ABs tested. The probability of identifying a phenotypically sensitive strain as resistant based on genotype is less than 5% for 85% of ABs.

According the overall estimations (Fig. 3) the 44% of the ABs, phenotypically resistant strains are genotypically associated with a 90% probability of resistance, while for the 92% of the ABs the phenotypically susceptible strains are genotypically susceptible with a 90%

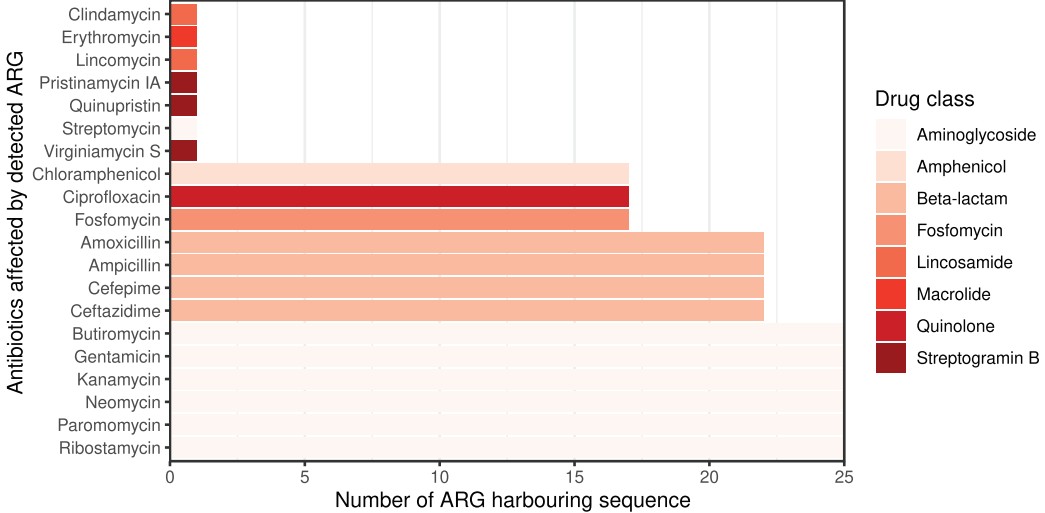

**Figure 4 Detected ARGs in the metagenome of the canine external otitis sample. The number of sequences harboring ARGs by affected AB compounds shown horizontally.** The color of the bars represents the drug class to which a given AB compound belongs.

probability. ARG detection showed a phenotypic prediction with at least 90% confidence in 67% of ABs. The probability of detecting a phenotypically susceptible strain as resistant based on genotype is below 5% for 92% of ABs. While the probability of detecting a phenotypically resistant strain as susceptible based on genotype is below 5% for 44% of ABs.

To have more detailed understanding of the agreement results, a further aspect of challenges related to AST should be considered. Even broth microdilution, which is the AST method proposed by EUCAST gives debatable results, and as do other classical, phenotype-based methods (*e.g.*, disk diffusion). Dissimilarity in laboratory conditions, human resources, and discrepancies in laboratory techniques may lead to different results (*Rebelo et al., 2022*).

A recent study including a machine learning-assisted ARG-based workflow for phenotypic AMR prediction indicates that the highest prediction accuracy for AMR against the drug classes tested appears by macrolides and sulfonamides. The highest uncertainty was found by beta-lactams that were also tested in combinations (*Hu et al., 2024*). From the demonstrated BioProjects, azithromycin, sulfamethoxazole, and sulfisoxazole fall into the above categories. In contrast to the above mentioned publication, only sulfixazole yielded above average results considering all the investigated factors (*NPV, PPV, SE, SP, ME, VME*). In line with the above-mentioned study, rather large deviations from the mean were found by beta-lactams. However, this finding be explained by the large number of beta-lactams. In our case 14 different compounds were tested in the above-mentioned drug class (ampicillin, aztreonam, cefepime, cefotaxime, cefoxitin, cefpodoxime, ceftazidime, ceftriaxone, cefuroxime, ertapenem, imipenem, meropenem, piperacillin, temocillin). These ABs belong to separate subcategories within beta-lactams

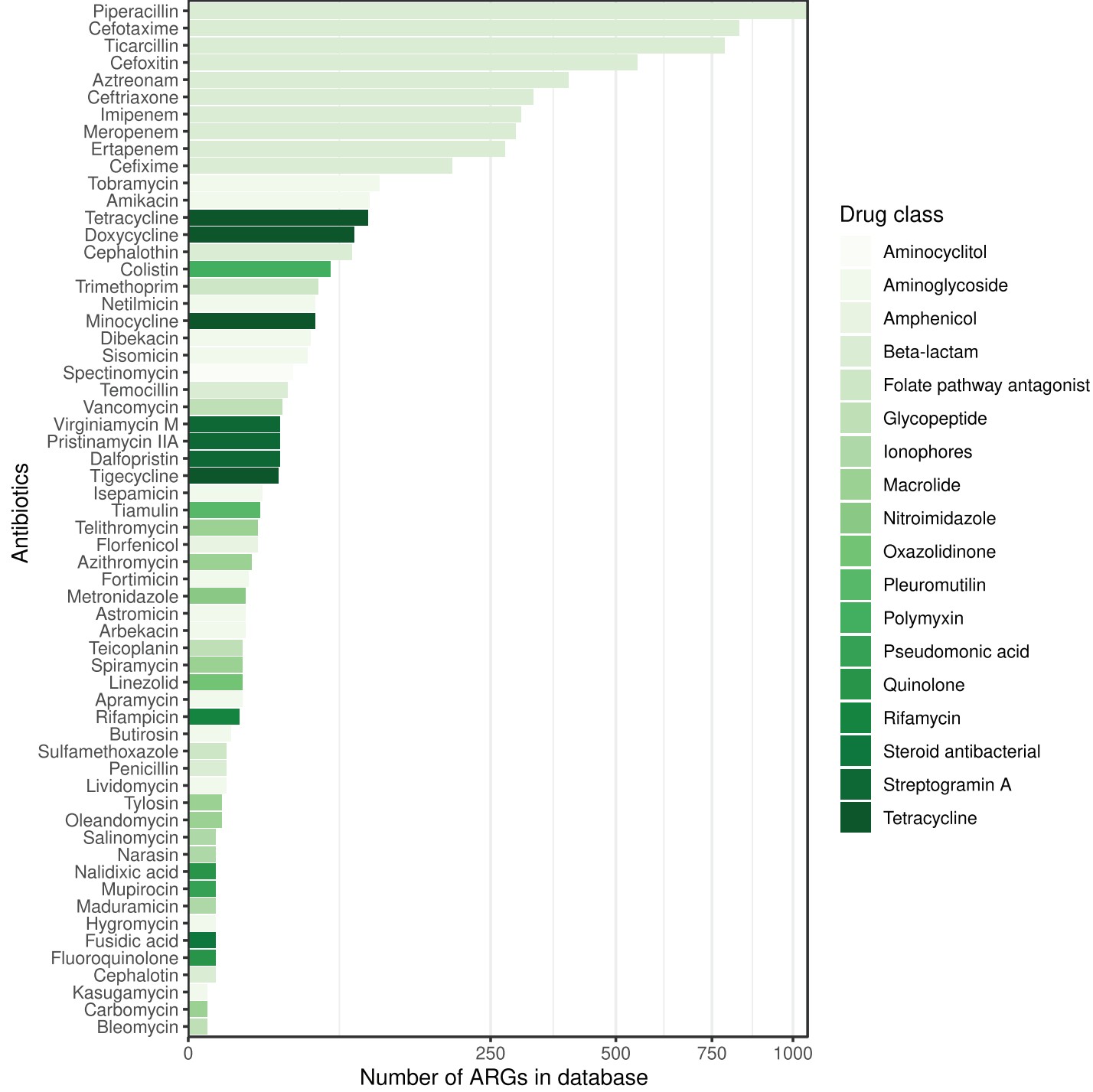

**Figure 5** **ABs for which no ARG was detected in the metagenomic sample are sorted by decreasing number of ARGs in the database affecting AB compounds.** The color of the bars represents the drug class to which a given AB compound belongs.

(cephalosporins, penams, penems, monobactams) that have very different properties from each other or even within the subcategory (*e.g.*, cephalosporin generations) (*Bush & Bradford, 2016*).

In contrast to single-strain cultures, similar comparisons using descriptive measures cannot theoretically be made with complex microbiological samples. One of the reasons for this is that only a small proportion of bacteria can be easily cultured and tested for phenotypic resistance (*Steen et al., 2019*). The recent advance of culturomics may partly address this aspect. Culturomics methods facilitates higher throughput, *i.e.*, the identification of numerous difficult-to-culture bacteria, by applying several different growth conditions and media simultaneously (*Lagier et al., 2015*). Even though, the number of condition combinations required for the same throughoutput can be experimentally reduced (*Diakite et al., 2020*), not all culturomics combinations are used in routine diagnostics. Thus, several bacteria involved in disease processes caused by a complex microbial communities can remain hidden, even if they play a critical role. However, shotgun metagenomic studies offer the opportunity to obtain data on the relative abundance rates of the taxa and, of particular clinical relevance, the genes, including ARGs of complex microbiomes. The ARG prevalence data can then be used to gain insight into the AMRP of a given sample. Unfortunately, due to the limitations of bacteriological culturing, the concordance of AMRP and complex bacterial phenotypic resistance cannot currently be depicted. However, if we think of the complex microbiological samples as sets of bacterial single-strains, we can assume that the phenotypic and genotypic resistance/susceptibility matches for the strains may also be extended to them.

The fact that the study was carried out on *E. coli* strains may also be taken into account when evaluating the results. According to the publication by *Hu et al. (2024)*, in which phenotypic AMR profile prediction was performed using genotype-based machine learning algorithms by various bacterial species, predictions for *E. coli* strains were less robust and reliable compared to other species such as *Campylobacter jejuni* and *Enterococcus faecium* (*Hu et al., 2024*). These differences in phenotypic AMR profile prediction at the bacterial species level may also be relevant in bacterial communities containing multiple taxonomic groups. Furthermore, in addition to AMR at the cellular level, there is an increasing focus on the so-called cooperative resistance of whole bacterial communities. The dynamics within microbial communities (regardless of being single-species populations or mixed populations) can influence the response to antimicrobial therapy (*Denk-Lobnig & Wood, 2023*). A bacterial cell that can inactivate ABs with an exoenzyme can confer immunity to a cell that does not have the enzyme (*Vega & Gore, 2014*). Such interspecies interactions are already known, *e.g. Stenotrophomonas maltophilia* can also degrade imipenem in an enzyme-mediated manner, thereby protecting susceptible *Pseudomonas aeruginosa* in cystic fibrosis (*Denk-Lobnig & Wood, 2023*). Cooperative resistance is present in all microbe-rich environments (*Vega & Gore, 2014*). Most clinical samples are microbe-rich and therefore using genotypic resistance testing in practice should be considered for its beneficial, additional value.

From a clinical point of view, the question is which ABs are effective against the bacteria involved in a complex microbial community that induces or sustains a disease process. From the presented studies of *E. coli* strains, phenotypic susceptibility can be predicted with high confidence for the vast majority of ABs tested based on the *ME*, *NPV*, *SP* metrics of the genotype. If we extend these approach to metagenomic datasets, its clinical applicability for the selection of therapeutic agents could be considered. Reflecting to the AMRP based on the metagenomic analysis presented above, we can conclude the following. Even infections caused by complex microbial communities can be associated with ABs that were not affected by any ARGs found, despite being associated with ARGs in the reference database. The number of ARGs for different ABs and drug classes may vary widely in the different databases, and it is important to note that each ARG database is incomplete. However, we can assume that ABs against which no ARGs can be detected in a sample are more likely to be clinically effective if the ARG reference databases include a relatively higher number of ARGs against them.

Conventional phenotypic AST procedures are still among the most significant methods for everyday routine bacteriological diagnostics. However, the adoption of ONT sequencing, followed by comprehensive bioinformatic analysis, allows for rapid detection of microorganisms, along with the assessment of their abundance rates, presence of various genes, including ARGs or even virulence genes, possibly all within a timeframe of 4 to 24 h (*Ring et al., 2023*). Consequently, clinical metagenomics based on ONT has the potential to enhance diagnostic microbiology and clinical practices significantly. Nonetheless, the innovative nature of this approach raises several questions that have been addressed within this study, including discrepancies observed between the identified ARGs and the expected phenotypic antimicrobial resistance (*Yee, Dien Bard & Simner, 2021*).

### Funding
The study was supported by the strategic research fund of the University of Veterinary Medicine Budapest (Grant No. SRF-001), and the European Union's Horizon 2020 research and innovation program supports the project under Grant Agreement No. 874735 (VEO). The funders had no role in study design, data collection and analysis, decision to publish, or preparation of the manuscript.

### Grant Disclosures
The following grant information was disclosed by the authors:
University of Veterinary Medicine Budapest: SRF-001.
European Union's Horizon 2020 research and innovation program supports: 874735 (VEO).

### Competing Interests
The authors declare that they have no competing interests.

## Author Contributions

- Norbert Solymosi conceived and designed the experiments, performed the experiments, analyzed the data, prepared figures and/or tables, authored or reviewed drafts of the article, and approved the final draft.
- Adrienn Gréta Tóth conceived and designed the experiments, performed the experiments, authored or reviewed drafts of the article, and approved the final draft.
- Sára Ágnes Nagy conceived and designed the experiments, authored or reviewed drafts of the article, and approved the final draft.
- István Csabai conceived and designed the experiments, authored or reviewed drafts of the article, and approved the final draft.
- Csongor Feczkó performed the experiments, authored or reviewed drafts of the article, and approved the final draft.
- Tamás Reibling conceived and designed the experiments, authored or reviewed drafts of the article, sample collection, and approved the final draft.
- Tibor Németh conceived and designed the experiments, performed the experiments, authored or reviewed drafts of the article, and approved the final draft.

## Animal Ethics

The following information was supplied relating to ethical approvals (*i.e.*, approving body and any reference numbers):

The sample we present is from a dog that was operated on with external otitis. In this operation, it is a routine procedure to collect specimens for microbiological control. This sampling is not an additional intervention; in this way, there is no requirement for research approval.

## DNA Deposition

The following information was supplied regarding the deposition of DNA sequences:

The data is available at NCBI BioProject: PRJNA1045271, PRJDB7087, PRJEB14086, PRJEB21880, PRJEB21997, PRJNA266657.

## Data Availability

The data is available at NCBI SRA: SRR31306311.

## Supplemental Information

Supplemental information for this article can be found online at http://dx.doi.org/10.7717/peerj.18802#supplemental-information.

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
