# Peer review of "Clinical considerations on antimicrobial resistance potential of complex microbiological samples"

_PeerJ, doi:10.7717/peerj.18802_

## Round 0.1 · original submission · Major Revisions

· Academic Editor

Major Revisions

Dear Dr. Norbert Solymosi,

Your manuscript presents valuable findings on microbial genomics and antimicrobial resistance (AMR), integrating both nanopore sequencing and Illumina data analysis. However, significant revisions are required to address reviewers' feedback and enhance the manuscript’s clarity, methodological rigor, and data transparency. Please address reviewers´ concerns, particularly the ones described below.

1. Clarification of Sequencing Technology and Study Focus
The manuscript centers on nanopore sequencing, but this focus is not clearly distinguished. It is essential to specify in the abstract and introduction that the study utilizes nanopore sequencing without a metagenomic approach. Providing context on the technology’s relevance, especially in clinical settings, will strengthen the manuscript’s impact.

2. Terminology on "Unculturable" Bacteria
Throughout the manuscript, "unculturable" bacteria is used inaccurately. As noted by reviewers, it is crucial to clarify that many bacteria remain uncultured due to unknown cultivation conditions, rather than being inherently "unculturable." Adjusting this terminology is essential for accurate scientific communication.

3. Data Accessibility
The manuscript currently states that raw long-read data are available upon request. To meet transparency standards, raw data should be deposited in a publicly accessible repository. This step will enhance reproducibility and align with data-sharing expectations in genomic research.

4. Sample Size and Experimental Details
In the Methods section, it is unclear how many samples were used for nanopore sequencing. Adding this information will help readers understand the study’s scope and robustness. Additionally, clarifying any statistical or control measures applied to account for sample variability will strengthen the manuscript’s experimental rigor.

5. Standardization of AMR Testing
The reviewers noted variability in AST standards across datasets. Standardizing the AST criteria used across different sources could improve comparability and consistency in AMR categorization, strengthening the study’s findings.

6. Intermediate Resistance and Multi-Drug Resistance Complexity
Including intermediate phenotypic resistance data would provide a more comprehensive picture of AMR, as intermediate resistance is clinically significant. Additionally, expanding the analysis to evaluate multi-drug resistance patterns, rather than single antibiotics alone, could offer insights into AMR complexity.

7. Typographical Corrections
All typographical issues noted in the PDF should be addressed. A thorough copy-edit will improve readability and presentation.

Conclusion
The manuscript has the potential to make a substantial contribution to AMR research, but major revisions are required to address reviewers' points and ensure methodological clarity, data accessibility, and accurate reporting. Implementing these changes will enhance the manuscript’s scientific rigor and impact.

Reviewer 1 ·

Basic reporting

The study involves nanopore sequencing, not any metagenomic approach. That should be clear from the very beginning, e.g. in the abstract, also introduction lacks the information on this particular technology and its use in clincal settings.

Throughout the manuscript, there's a misunderstanding of "can not be cultured" bacteria. We indeed can not culture a lot of bacteria but that does not mean that they are not culturable. It means that we still do not know how to culture them.

Raw data unavailable. This is not acceptable: "The raw long-read data are available from the corresponding author upon reasonable request.".

Typos pointe out in the pdf.

Experimental design

In the Methods and in general, it is not clear how many samples were taken for Nanopore sequencing.

Validity of the findings

no comment

Annotated reviews are not available for download in order to protect the identity of reviewers who chose to remain anonymous.

·

Basic reporting

This study presents a comprehensive approach to investigating phenotypic and genotypic antimicrobial resistance (AMR) by combining antibiograms and Illumina sequencing data from E. coli strains.

Experimental design

The use of multiple BioProjects ensuring a dataset that allows for significant cross-comparisons. Additionally, the bioinformatic pipeline, including quality control with TrimGalore, assembly using MEGAHIT, and ARG detection through ResFinder. The statistical analyses employing 2x2 cross-tabulations and key metrics like sensitivity, specificity, PPV, and NPV to assess the predictive power of genotypic methods against phenotypic AMR data.

Validity of the findings

There are few notable weaknesses. First, the variation in AST standards and cut-off versions across different BioProjects may introduce inconsistency in resistance categorization. This could impact the comparison of results across datasets. Second, the exclusion of intermediate phenotypic results limits the completeness of the study, as intermediate resistance is often clinically significant. Additionally, the focus on single antibiotics rather than combinations may oversimplify the complexity of multi-drug resistance patterns.

Additional comments

To improve the study, following may be considered:
1. Standardizing the AST criteria across BioProjects to reduce variability in resistance classifications. 2. Including intermediate resistance data in analysis may provide a more comprehensive understanding of AMR.
3. Expanding the scope to assess antibiotic combinations could offer more insight into multi-drug resistance, a critical aspect of AMR research.

---

## Round 0.2 · Minor Revisions

· Academic Editor

Minor Revisions

Dear Dr. Solymosi,,

Please check the comments for the text improvement in the graphical abstract before the paper can be accepted.

Reviewer 1 ·

Basic reporting

Almost all highlighted points have been revised and improved. The authors probably unintentionally skipped my comments for the text improvement in the graphical abstract. **Please recheck this**

Experimental design

To the authors: Thank you for clarification and improvements in the text.

Validity of the findings

no comment

Additional comments

no comment

·

Basic reporting

The article is now clear, professionally written, well-structured, and self-contained, with sufficient context, literature references, relevant results, figures, tables, and shared raw data.

Experimental design

The manuscript presents original research aligned with the journal's aims, addressing a well-defined knowledge gap through rigorous investigation, detailed methods, and self-contained results.

Validity of the findings

The study is robust, well-supported, self-contained, and its conclusions are appropriately tied to the research question, though replication with clear rationale is encouraged.

---

## Round 0.3 · accepted · Accept

· Academic Editor

Accept

Congratulations for the acceptance of your manuscript,

Best regards
Rodrigo